# COVID-19 Vaccine Hesitancy Worldwide: A Concise Systematic Review of Vaccine Acceptance Rates

**DOI:** 10.3390/vaccines9020160

**Published:** 2021-02-16

**Authors:** Malik Sallam

**Affiliations:** 1Department of Pathology, Microbiology and Forensic Medicine, School of Medicine, The University of Jordan, Amman 11942, Jordan; malik.sallam@ju.edu.jo; Tel.: +962-79-184-5186; 2Department of Clinical Laboratories and Forensic Medicine, Jordan University Hospital, Amman 11942, Jordan

**Keywords:** vaccine hesitancy, vaccine acceptance, anti-vaccination, vaccination coverage rates, COVID-19, coronavirus, SARS-CoV-2, vaccine rejection

## Abstract

Utility of vaccine campaigns to control coronavirus 2019 disease (COVID-19) is not merely dependent on vaccine efficacy and safety. Vaccine acceptance among the general public and healthcare workers appears to have a decisive role in the successful control of the pandemic. The aim of this review was to provide an up-to-date assessment of COVID-19 vaccination acceptance rates worldwide. A systematic search of the peer-reviewed English survey literature indexed in PubMed was done on 25 December 2020. Results from 31 peer-reviewed published studies met the inclusion criteria and formed the basis for the final COVID-19 vaccine acceptance estimates. Survey studies on COVID-19 vaccine acceptance rates were found from 33 different countries. Among adults representing the general public, the highest COVID-19 vaccine acceptance rates were found in Ecuador (97.0%), Malaysia (94.3%), Indonesia (93.3%) and China (91.3%). However, the lowest COVID-19 vaccine acceptance rates were found in Kuwait (23.6%), Jordan (28.4%), Italy (53.7), Russia (54.9%), Poland (56.3%), US (56.9%), and France (58.9%). Only eight surveys among healthcare workers (doctors and nurses) were found, with vaccine acceptance rates ranging from 27.7% in the Democratic Republic of the Congo to 78.1% in Israel. In the majority of survey studies among the general public stratified per country (29/47, 62%), the acceptance of COVID-19 vaccination showed a level of ≥70%. Low rates of COVID-19 vaccine acceptance were reported in the Middle East, Russia, Africa and several European countries. This could represent a major problem in the global efforts to control the current COVID-19 pandemic. More studies are recommended to address the scope of COVID-19 vaccine hesitancy. Such studies are particularly needed in the Middle East and North Africa, Sub-Saharan Africa, Eastern Europe, Central Asia, Middle and South America. Addressing the scope of COVID-19 vaccine hesitancy in various countries is recommended as an initial step for building trust in COVID-19 vaccination efforts.

## 1. Introduction

Based on the Strategic Advisory Group of Experts on Immunization (SAGE), vaccine hesitancy is the term used to describe: “delay in acceptance or refusal of vaccination despite availability of vaccination services” [1]. Factors that affect the attitude towards acceptance of vaccination include complacency, convenience and confidence [1,2]. Complacency denotes the low perception of the disease risk; hence, vaccination was deemed unnecessary. Confidence refers to the trust in vaccination safety, effectiveness, besides the competence of the healthcare systems. Convenience entails the availability, affordability and delivery of vaccines in a comfortable context [2].

The complex nature of motives behind vaccine hesitancy can be analysed using the epidemiologic triad of environmental, agent and host factors [3,4]. Environmental factors include public health policies, social factors and the messages spread by the media [5,6,7]. The agent (vaccine and disease) factors involve the perception of vaccine safety and effectiveness, besides the perceived susceptibility to the disease [7,8,9]. Host factors are dependent on knowledge, previous experience, educational and income levels [4,10].

Previous studies have shown that vaccine hesitancy is a common phenomenon globally, with variability in the cited reasons behind refusal of vaccine acceptance [11,12,13]. The most common reasons included: perceived risks vs. benefits, certain religious beliefs and lack of knowledge and awareness [14,15,16]. The aforementioned reasons can be applied to COVID-19 vaccine hesitancy, as shown by the recent publications that showed a strong correlation between intent to get coronavirus vaccines and its perceived safety [17], association of the negative attitude towards COVID-19 vaccines and unwillingness to get the vaccines [18], and the association of religiosity with lower intention to get COVID-19 vaccines [19].

Studying the global impact of vaccine hesitancy—including willingness to accept COVID-19 vaccines—could be complicated by the multifaceted nature of this phenomenon [1]. This entails the existence of cognitive, psychologic, socio-demographic and cultural factors that contribute to vaccine hesitancy [20,21,22,23]. Analysis of such factors is needed to address COVID-19 vaccine hesitancy, following the assessment of the scope and magnitude of this public health threat [24]. This can help in guiding interventional measures aimed at building and maintaining responses to tackle this threat [25].

Earlier studies that assessed attitudes towards vaccines revealed the existence of regional variability in perceiving the safety and effectiveness of vaccination [12,26,27]. Higher-income regions were the least certain regarding vaccine safety with 72%–73% of people in Northern America and Northern Europe who agreed that vaccines are safe. This rate was even lower in Western Europe (59%), and in Eastern Europe (50%), despite the presence of a substantial variability in Eastern European countries (from 32% in Ukraine, 48% in Russia, to 77% in Slovakia). However, the majority of people in lower-income areas agreed that vaccines are safe, with the highest proportions seen in South Asia (95%) and in Eastern Africa (92%) [26]. A similar pattern was observed regarding vaccine effectiveness, with Eastern Europe as the region where people are the least likely to agree that vaccines are effective, as opposed to South Asia and Eastern Africa [26]. The assessment of such regional differences can be invaluable in addressing and fighting public health threats posed by vaccine hesitancy [28].

The current coronavirus disease 2019 (COVID-19) pandemic does not seem to show any signs of decline, with more than 1.7 million deaths and more than 80 million reported cases worldwide, as of 27 December 2020 [29,30]. The ebb and flow of COVID-19 cases can be driven by human factors, including attitude towards physical distancing and protective measures, while viral factors are driven by mutations that commonly occur in severe acute respiratory syndrome Coronavirus 2 (SARS-CoV-2) genome [31,32,33,34,35,36,37]. The viral factors can particularly be of high relevance considering the recent reports of resurgence in COVID-19 infections in UK due to a new variant of the virus [38].

The global efforts to lessen the effects of the pandemic, and to reduce its health and socio-economic impact, rely to a large extent on the preventive efforts [39,40]. Thus, huge efforts by the scientific community and pharmaceutical industry backed by governments’ support, were directed towards developing efficacious and safe vaccines for SARS-CoV-2 [41]. These efforts were manifested by the approval of several vaccines for emergency use, in addition to more than 60 vaccine candidates in clinical trials. Moreover, more than 170 COVID-19 vaccine candidates are in the pre-clinical phase [42].

Despite the huge efforts made to achieve successful COVID-19 vaccines, a major hindrance can be related to vaccine hesitancy towards the approved and prospective COVID-19 vaccination [43]. To identify the scope of this problem, this systematic review aimed to assess the acceptance rates for COVID-19 vaccine(s) in different countries worldwide, which can provide an initial step to study the factors implicated in regional and cultural differences behind COVID-19 vaccine hesitancy.

## 2. Materials and Methods

This review was conducted following the PRISMA guidelines [44].

Published papers in PubMed/Medline that aimed at evaluating COVID-19 vaccine hesitancy/vaccine acceptance using a survey/questionnaire were eligible for inclusion in this review.

Only studies in English language that met the inclusion criteria were considered in this review. The inclusion criteria were: (1) peer-reviewed published articles indexed in PubMed; (2) survey studies among the general population, health-care workers, students, or parents/guardians); (3) the major aim of the study was to evaluate COVID-19 vaccine acceptance/hesitancy; and (4) publication language was English.

The exclusion criteria were: (1) unpublished manuscripts (preprints); (2) the article did not aim to evaluate COVID-19 vaccine acceptance/hesitancy; and (3) publication language was not English.

A search was done as of 25 December 2020, using the following strategy: (COVID * vaccine * hesitancy [Title/Abstract]) OR (COVID * vaccine acceptance[Title/Abstract])) OR (COVID * vaccine * hesitanc *[Title/Abstract])) OR (COVID * intention to vaccine * [Title/Abstract]) OR (COVID* vaccine * accept *[Title/Abstract]) AND (2020:2020[pdat]).

Screening of titles and abstracts was conducted, followed by data extraction for the following items: date of survey, country/countries in which the survey was conducted, target population for survey (e.g., general public, healthcare workers, and students), total number of respondents, and COVID-19 vaccine acceptance rate (which included the number of respondents who answered: agree/somewhat/completely agree/leaning towards yes/definitely yes).

## 3. Results

A total of 178 records were identified, and following the screening process, a total of 30 articles were included in this review (Figure 1). In addition, data collected in a recently published article that surveyed the general public residing in Jordan and Kuwait were added to the final analysis [45].

### 3.1. Characteristics of the Papers Included in This Review

A total of 30 published papers were analysed in this review, with an additional recently published article that focused on COVID-19 vaccine acceptance in Jordan and Kuwait to yield a total of 31 studies. These studies comprised surveys on COVID-19 vaccine acceptance from a total of 33 different countries. Surveys were done most commonly in the UK (*n* = 6), followed by France and the US (*n* = 5, for each country), and China and Italy (*n* = 4, for each country). Dates of survey distribution ranged from February 2020 until December 2020. A few studies were conducted in more than one country, including the study by Lazarus et al., involving 19 countries and the study by Neumann-Böhme et al., involving seven European countries [46,47].

Stratified per country, a total of 60 surveys were found with the largest sample size (*n* = 5114) in the study conducted in the UK by Freeman et al., while the smallest sample size (*n* = 123) was found in the study conducted in Malta by Gretch et al., among general practitioners and trainees [48,49]. Out of these 60 surveys, 47 were among the general public, eight surveys were among healthcare workers (doctors, nurses, or others), three surveys were among parents/guardians and two surveys involved University students (Table 1). Surveys were most commonly conducted in June or July (23/60, 38%), followed by March or April (20/60, 33%).

### 3.2. Rates of COVID-19 Vaccine Acceptance

The results of the COVID-19 vaccine acceptance rates in different studies included in this review and stratified by country are shown in Table 1. Classified per study, the highest vaccine acceptance rates (>90%) among the general public were found in four studies from Ecuador (97.0%), Malaysia (94.3%), Indonesia (93.3%) and China (91.3%). On the contrary, the lowest vaccine acceptance rates (<60%) among the general public were found in seven studies to be from Kuwait (23.6%), Jordan (28.4%), Italy (53.7), Russia (54.9%), Poland (56.3%), US (56.9%), and France (58.9%). In Figure 2, COVID-19 vaccine acceptance rates are shown per country, with the latest estimate used for countries with multiple studies.

For the eight studies conducted on healthcare workers, three surveys reported vaccine acceptance rates below 60%, with the highest rate being among doctors in Israel (78.1%) and the lowest vaccine acceptance rate (27.7%) reported among healthcare workers in the Democratic Republic of the Congo (DRC).

For the three studies conducted among parents/guardians, the vaccine acceptance rates were more than 70%. For the two studies among University students, the vaccine acceptance rate was 57.3% in Malta (excluding university staff), and 86.1% in Italy.

Male sex was associated with significantly higher rates of COVID-19 vaccine in 15 countries/studies, while the age was a significant factor in 11 studies/countries.

### 3.3. Changes in COVID-19 Vaccine Acceptance over Time in Countries with Multiple Survey Studies

In countries with multiple surveys over time, the following changes in COVID-19 vaccine acceptance rates were observed. In the UK, the vaccine acceptance rate was 79.0% in April, 83.0% in May, 71.5% in June, 64.0% in July and 71.7% in September/October. In France, the vaccine acceptance rate ranged from 62.0% to 77.1% in March/April and was 58.9% in June. In Italy, the vaccine acceptance rate was 77.3% in April, 70.8% in June and it reached 53.7% in September.

For the vaccine acceptance rates in the US, it was 56.9% in April, and ranged from 67.0% to 75.0% in May, and reached 75.4% in June. In China, three studies reported high rates of vaccine acceptance with the first study that reported a vaccine acceptance rate of 91.3% in March, the second study reported a rate of 83.5% in May and the third study reported a rate of 88.6% in June.

## 4. Discussion

Vaccine hesitancy is an old phenomenon that represents a serious threat to the global health, as shown by the resurgence of some infectious diseases (e.g., outbreaks of measles and pertussis) [76,77,78,79,80]. The huge leaps in developing efficacious and safe COVID-19 vaccines within a short period were unprecedented [81,82,83]. Nevertheless, COVID-19 vaccine hesitancy can be the limiting step in the global efforts to control the current pandemic with its negative health and socio-economic effects [43,84,85].

Assessing the level of population immunity necessary to limit the pathogen spread is dependent on the basic reproductive number for that infectious disease [86]. The latest estimates on COVID-19, pointed out a range of 60–75% immune individuals that would be necessary to halt the forward transmission of the virus and community spread of the virus [87,88,89]. Vaccine cost, effectiveness and duration of protection appear as important factors to achieve such a goal [83,90,91]. However, vaccine hesitancy can be a decisive factor that would hinder the successful control of the current COVID-19 pandemic [43,92]. Thus, estimates of vaccine acceptance rates can be helpful to plan actions and intervention measures necessary to increase the awareness and assure people about the safety and benefits of vaccines, which in turn would help to control virus spread and alleviate the negative effects of this unprecedented pandemic [93,94]. Evaluation of attitudes and acceptance rates towards COVID-19 vaccines can help to initiate communication campaigns that are much needed to strengthen trust in health authorities [24].

In this review, a large variability in COVID-19 vaccine acceptance rates was found. However, certain patterns can be deduced based on descriptive analysis of the reported vaccine acceptance rates. First, in East and South East Asia, the overall acceptance rates among the general public were relatively high. This includes more than 90% acceptance rates in Indonesia, Malaysia and one study from China [51,52,59]. Another two surveys on the general public in China reported vaccine acceptance rates of more than 80%, with an additional survey in South Korea that reported a rate of 79.8% [46,63]. A later survey from Shenzhen, China, by Zhang et al., which surveyed parents/guardians who were factory workers, on their acceptability of children COVID-19 vaccination reported a lower rate of 72.5% compared to previous studies [71]. Similarly, an online survey on Australian parents showed an acceptance rate of 75.8%, dropping from a rate of 85.8% in April among adults in Australia who were surveyed in April 2020 [68,95]. The lowest COVID-19 vaccine acceptance rate among the general public in the region was reported by Lazarus et al., in Singapore (67.9%) [46]. The relatively high rates of vaccine acceptance in the region were attributed to strong trust in governments [46]. Additionally, the only survey in India reported a vaccine acceptance rate of 74.5% [46]. The relatively high rates of COVID-19 vaccine acceptance might be related to stronger confidence in vaccine safety and effectiveness, as reported previously in Asia [27].

However, two studies that dated back to the early part of the pandemic (February and March) among nurses in Hong Kong reported low rates of COVID-19 acceptance (40.0% and 63.0%) [50,55]. Likewise, Kabamba Nzaji et al. reported a very low rate of COVID-19 vaccine acceptance among healthcare workers in the DRC (27.7%) [56]. This issue is alarming considering the front-line position of healthcare workers in fighting the spread and effects of the COVID-19 pandemic, which put them at a higher risk of infection, and hence their higher need for protective measures [96,97,98].

Additionally, the vaccine acceptance rates were relatively high in Latin America, where results from Brazil and Ecuador reported more than 70% acceptance rates [46,58]. This was also seen in the survey from Mexico with a vaccine acceptance rate of 76.3% [46].

In Europe, the results were largely variable, with countries around the Mediterranean reporting vaccine acceptance rates as low as 53.7% in Italy, and 58.9% in France; no surveys among the general public in Malta were found [46,72]. The results in Italy and France can be viewed from the perspective of lacking confidence in the safety of these vaccines, since such a negative attitude was reported previously in these countries [27]. In addition, low rates of COVID-19 vaccine acceptance were reported among students and healthcare workers in Malta—44.2% and 52.0%, respectively [73,74]. Variable results were also reported in other European countries with rates as high as 80.0% in Denmark, and as low as 56.3% in Poland [46,47]. The vaccine acceptance rates were even lower in Russia (54.9%), which needs further evaluation considering the heavy toll of COVID-19 on the country [29,46]. Variability in vaccine acceptance rates was also seen in the UK, US and Canada over the course of the pandemic [61,62,64,65,70]. Additionally, a drop in COVID-19 vaccine acceptance was noticed in a few European countries, which is in line with the recent report by Lin et al. [24]. Such patterns of COVID-19 vaccine hesitancy were consistent with a previous report that showed relatively high rates of vaccine hesitancy in Western and Eastern Europe, in addition to Russia [26]. The aforementioned low rates can be linked to lower confidence in vaccine safety and effectiveness in these regions [26].

The Middle East was among the regions with the lowest COVID-19 vaccine acceptance rates globally. The acceptance rate was the lowest in Kuwait (23.6%), followed by Jordan (28.4%), Saudi Arabia (64.7%) and Turkey (66.0%) [45,62,75]. Such low rates can be related to the widespread embrace of conspiratorial beliefs in the region, with its subsequent negative attitude towards vaccination [23,99,100,101]. However, the highest vaccine acceptance rate was reported in Israel (75.0%); however, this rate was much lower among nurses surveyed in the same study (61.1%) [53].

Only two surveys among the general public in African countries reported an acceptance rate of 81.6% in South Africa and 65.2% in Nigeria [46]. Early knowledge, attitudes and practices survey study towards COVID-19, from North-Central Nigeria, reported an acceptance rate of barely 29.0%, which highlights the need for more studies for an accurate depiction of COVID-19 vaccine hesitancy in Africa due to possible large regional and sub-regional variations [102]. Thus, more studies are recommended in Africa to address COVID-19 vaccine hesitancy in the continent. Despite the previous findings of an overall low prevalence of vaccine hesitancy in Eastern Africa, the attitude towards the newer vaccines, including those of COVID-19, remains a study topic that has not been explored to a large degree [12]. Besides Africa, more studies are needed from Central Asia, Eastern Europe, Central and South America to reach reliable conclusions about the scope of COVID-19 vaccine hesitancy around the globe.

Finally, the assessment of the role of sex and age in COVID-19 vaccine hesitancy revealed that males were more inclined to accept COVID-19 vaccines. This can be related to their higher perception of COVID-19 dangers and lower belief in conspiratorial claims surrounding the disease [45,99,101]. These variables should be considered for an accurate interpretation of COVID-19 acceptance rates, since sampling bias, particularly in sex distribution, can affect the reported rates.

The limitations of this review include the sole dependence on PubMed in the search study; however, this approach was done to provide a concise and succinct evaluation of COVID-19 vaccine hesitancy. This approach could have resulted in the inevitable missing of a few relevant studies tackling the subject of this review (e.g., the study by Head et al. assessing SARS-CoV-2 vaccination intentions among adults in the US) [103]. In addition, the research studies included in this review represented cross-sectional studies, which can be seen as snapshots of vaccine hesitancy status in each country/region, with different sampling strategies, which may partly explain the differences in vaccine acceptance rates reported in various studies from a single country. Thus, the results should be interpreted with extreme caution since they cannot predict the future changes in vaccine acceptance rates. The results of this study can be used as an initial motivation and guide for future studies and vaccine awareness campaigns. Finally, an important limitation was related to the different approaches used to express the willingness to accept COVID-19 vaccines in various studies (i.e., some studies used a binary response of yes/no, while others used a scale of strongly agree/agree/neutral/disagree/strongly disagree to deduce the inclination towards vaccine acceptance, etc.); thus, this variable should be taken into account for accurate comparisons of vaccine acceptance rates between different studies.

## 5. Conclusions

Large variability in COVID-19 vaccine acceptance rates was reported in different countries and regions of the world. A sizable number of studies reported COVID-19 acceptance rates below 60%, which would pose a serious problem for efforts to control the current COVID-19 pandemic. Low COVID-19 vaccine acceptance rates were more pronounced in the Middle East, Eastern Europe and Russia. High acceptance rates in East and South East Asia would help to achieve proper control of the pandemic. More studies are recommended to assess the attitude of general public and healthcare workers in Africa, Central Asia and the Middle East besides Central and South America. Such studies would help to evaluate COVID-19 vaccine hesitancy and its potential consequences in these regions, and around the globe.

The major challenges that could face successful implementation of COVID-19 vaccination programs to fight the unprecedented pandemic include mass manufacturing of vaccines, its fair distribution across the world and the uncertainty regarding its long-term efficacy. However, vaccine hesitancy can be the major hindrance of the control efforts to lessen the negative consequences of COVID-19 pandemic, at least in certain countries/regions.

The widespread prevalence of COVID-19 vaccine hesitancy mandates collaborative efforts of governments, health policy makers, and media sources, including social media companies. It is recommended to build COVID-19 vaccination trust among the general public, via the spread of timely and clear messages through trusted channels advocating the safety and efficacy of currently available COVID-19 vaccines.

## Figures and Tables

**Figure 1 vaccines-09-00160-f001:**
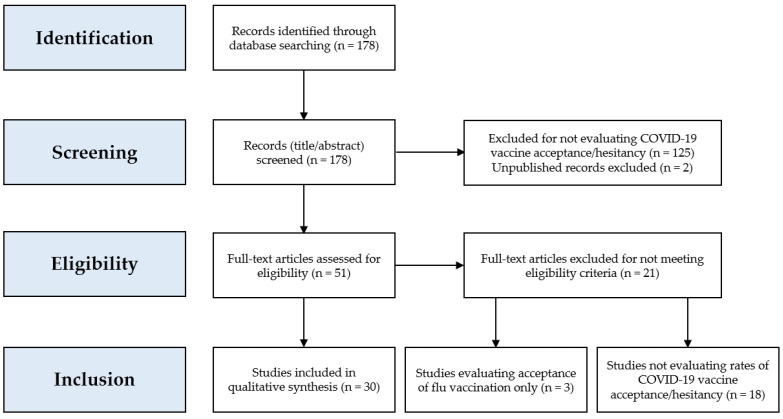
Flow chart of the study selection process.

**Figure 2 vaccines-09-00160-f002:**
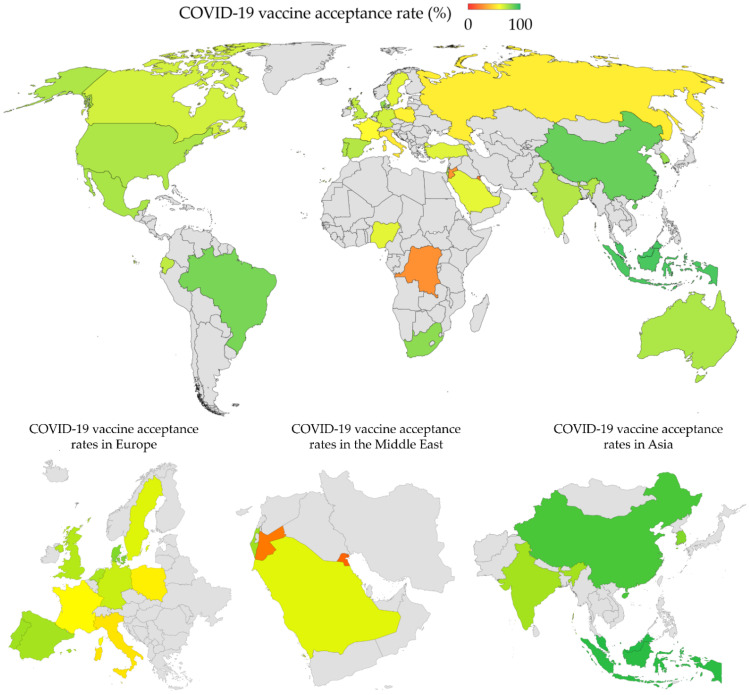
COVID-19 vaccine acceptance rates worldwide. For countries with more than one survey study, the vaccine acceptance rate of the latest survey was used in this graph. The estimates were also based on studies from the general population, except in the following cases where no studies from the general public were found (Australia: parents/guardians; DRC: healthcare workers; Hong Kong: healthcare workers; Malta: healthcare workers).

**Table 1 vaccines-09-00160-t001:** COVID-19 acceptance rates divided by the included studies and sorted based on the date of survey.

Study	Country	Date of Survey	Response Recorded as Vaccine Acceptance	N ^6^	Target Population	Acceptance Rate (%)	Age/Sex Correlation with Higher Vaccine Acceptance
Wang et al. [50]	Hong Kong	February and March, 2020	Intend to accept	806	Nurses	40.0	Male
Wang et al. [51]	China	March, 2020	Yes	2058	General population	91.3	Male
Harapan et al. [52]	Indonesia	March and April 2020	Yes	1359	General population	93.3	None
Dror et al. [53]	Israel	March and April 2020	Yes	388	Doctors	78.1	-
Detoc et al. [54]	France	March and April 2020	Yes certainly/possibly	3259	General population	77.6	Male, age
Dror et al. [53]	Israel	March and April 2020	Yes	1112	General population	75.0	-
Kwok et al. [55]	Hong Kong	March and April 2020	Likely to vaccinate (scored 6 or above out of 10)	1205	Nurses	63.0	Age
Dror et al. [53]	Israel	March and April 2020	Yes	211	Nurses	61.1	-
Nzaji et al. [56]	DRC ^2^	March and April 2020	Yes	613	Healthcare workers	27.7	Age
Gagneux-Brunon et al. [57]	France	March to July, 2020	Yes	2047	Healthcare workers	76.9	Male, age
Sarasty et al. [58]	Ecuador	April, 2020	Willing to accept a vaccine	1050	General population	97.0	-
Wong et al. [59]	Malaysia	April, 2020	Definitely, probably or possibly yes	1159	General population	94.3	Male
Neumann-Böhme et al. [47] ^1^	Denmark	April, 2020	Yes	1000	General population	80.0	-
Neumann-Böhme et al. [47]	UK ^3^	April, 2020	Yes	1000	General population	79.0	-
Neumann-Böhme et al. [47]	Italy	April, 2020	Yes	1500	General population	77.3	-
Ward et al. [60]	France	April and May 2020	Certainly or probably	5018	General population	76.0	None
Neumann-Böhme et al. [47]	Portugal	April, 2020	Yes	1000	General population	75.0	-
Neumann-Böhme et al. [47]	Netherland	April, 2020	Yes	1000	General population	73.0	-
Neumann-Böhme et al. [47]	Germany	April, 2020	Yes	1000	General population	70.0	-
Neumann-Böhme et al. [47]	France	April, 2020	Yes	1000	General population	62.0	-
Fisher et al. [61]	US ^4^	April, 2020	Yes	1003	General population	56.9	Male, age
Salali & Uysal [62]	UK	May, 2020	Yes	1088	General population	83.0	None
Lin et al. [63]	China	May, 2020	Definitely/probably yes	3541	General population	83.5	None
Taylor et al. [64]	Canada	May, 2020	Yes	1902	General population	80.0	Male, age
Taylor et al. [64]	US	May, 2020	Yes	1772	General population	75.0	Male, age
Salali & Uysal [62]	Turkey	May, 2020	Yes	3936	General population	66.0	Male
Reiter et al. [65]	US	May, 2020	Definitely/probably willing	2006	General population	68.5	Male
Malik et al. [66]	US	May, 2020	Agree/strongly agree	672	General population	67.0	Male, age
Lazarus et al. [46]	China	June, 2020	Completely/somewhat agree	712	General population	88.6	-
Barello et al. [67]	Italy	June, 2020	Yes	735	University students	86.1	-
Lazarus et al. [46]	Brazil	June, 2020	Completely/somewhat agree	717	General population	85.4	-
Lazarus et al. [46]	South Africa	June, 2020	Completely/somewhat agree	619	General population	81.6	-
Lazarus et al. [46]	South Korea	June, 2020	Completely/somewhat agree	752	General population	79.8	-
Lazarus et al. [46]	Mexico	June, 2020	Completely/somewhat agree	699	General population	76.3	-
Lazarus et al. [46]	US	June, 2020	Completely/somewhat agree	773	General population	75.4	-
Lazarus et al. [46]	India	June, 2020	Completely/somewhat agree	742	General population	74.5	-
Lazarus et al. [46]	Spain	June, 2020	Completely/somewhat agree	748	General population	74.3	-
Lazarus et al. [46]	Ecuador	June, 2020	Completely/somewhat agree	741	General population	71.9	-
Lazarus et al. [46]	UK	June, 2020	Completely/somewhat agree	768	General population	71.5	-
Lazarus et al. [46]	Italy	June, 2020	Completely/somewhat agree	736	General population	70.8	-
Lazarus et al. [46]	Canada	June, 2020	Completely/somewhat agree	707	General population	68.7	-
Lazarus et al. [46]	Germany	June, 2020	Completely/somewhat agree	722	General population	68.4	-
Lazarus et al. [46]	Singapore	June, 2020	Completely/somewhat agree	655	General population	67.9	-
Lazarus et al. [46]	Sweden	June, 2020	Completely/somewhat agree	650	General population	65.2	-
Lazarus et al. [46]	Nigeria	June, 2020	Completely/somewhat agree	670	General population	65.2	-
Lazarus et al. [46]	France	June, 2020	Completely/somewhat agree	669	General population	58.9	-
Lazarus et al. [46]	Poland	June, 2020	Completely/somewhat agree	666	General population	56.3	-
Lazarus et al. [46]	Russia	June, 2020	Completely/somewhat agree	680	General population	54.9	-
Rhodes et al. [68]	Australia	June, 2020	Yes	2018	Parents and guardians	75.8	Male, age
Bell et al. [69]	UK	July, 2020	Yes, definitely or unsure but leaning towards yes	1252	Parents and guardians	89.1	-
Sherman et al. [70]	UK	July, 2020	Very likely	1500	General population	64.0	Age
Zhang et al. [71]	China	September, 2020	Likely or very likely	1052	Parents and guardians	72.6	None
Gretch et al. [49]	Malta	September, 2020	Likely	123	GPs and GP trainees	61.8	-
La Vecchia et al. [72]	Italy	September, 2020	Yes/probably yes	1055	General population	53.7	-
Gretch et al. [73]	Malta	September, 2020	Likely	1002	Healthcare workers	52.0	-
Gretch & Gauci [74]	Malta	September, 2020	Likely	852	University students/staff	44.2	-
Freeman et al. [48]	UK	September and October, 2020	Endorsing 4/7 items of Oxford Scale ^5^	5114	General population	71.7	Male, age
Al-Mohaithef & Badhi [75]	Saudi Arabia	Unknown	Yes	992	General population	64.7	None
Sallam et al. [45]	Jordan	December, 2020	Yes	2173	General population	28.4	Male
Sallam et al. [45]	Kuwait	December, 2020	Yes	771	General population	23.6	Male

^1^ The study by Neumann-Böhme et al. reported that males and participants > 55 years were more willing to accept COVID-19 vaccines; however, this finding was not stratified per country. ^2^ DRC: The Democratic Republic of the Congo; ^3^ UK: United Kingdom; ^4^ US: United States; ^5^ Oxford Scale: Oxford COVID-19 vaccine hesitancy scale developed by Freeman et al. [48]; ^6^ N: Number.

## Data Availability

Data supporting this systematic review are available in the reference section. In addition, the analyzed data that were used during the current systematic review are available from the author on reasonable request.

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
