# Peer review of "COVID-19 Vaccine Hesitancy Worldwide: A Concise Systematic Review of Vaccine Acceptance Rates"

_vaccines, 2021, doi:10.3390/vaccines9020160_

Round 1

Reviewer 1 Report

Very interesting study.

The conclusions section needs to be more focused on how national and international public health institutions can use the evidence from the study to manage vaccination hesitation

Author Response

  1. Very interesting study.

The conclusions section needs to be more focused on how national and international public health institutions can use the evidence from the study to manage vaccination hesitation

Response: I am deeply grateful for the reviewer comment, and based on the suggestion made by the reviewer, the following paragraphs were added to the conclusions section:

(Page: 10, lines: 320-330)

“The major challenges that could face successful implementation of COVID-19 vaccination programs to fight the unprecedented pandemic include mass manufacturing of vaccines, its fair distribution across the world and the uncertainty regarding its long-term efficacy. However, vaccine hesitancy can be the major hindrance of the control efforts to lessen the negative consequences of COVID-19 pandemic, at least in certain countries/regions.

The widespread prevalence of COVID-19 vaccine hesitancy mandates collaborative efforts of government, health policy makers, and media sources including social media companies. It is recommended to build COVID-19 vaccination trust among the general public, via the spread of timely and clear messages through trusted channels advocating the safety and efficacy of currently available COVID-19 vaccines.”

Reviewer 2 Report

This study aims to compare the COVID-19 vaccine acceptance rates across countries by conducting a literature review on the recent survey studies conducted around the world.

I think this is an important review as it will allow researchers to observe the cross-cultural differences in COVID-19 vaccine acceptance rates and to investigate further why people in some countries are more hesitant about vaccinations compared to others. Therefore, I would like to see this article published. I have some concerns about the methodology, that I think we need further information on the article screening process to ensure the literature review was thorough and all eligible articles were included. Once we have more information on the eligibility criteria and an extended discussion and introduction (see details below), I think the manuscript will be suitable for publication.

Here are my further comments:

Introduction: I would like to see more extended background information on the cross-cultural differences on vaccine hesitancy so that the author can also make predictions about which countries we would expect the COVID-19 vaccine acceptance rates would be lower. There are a few cross-cultural studies previously conducted on vaccine hesitancy worldwide, which should be cited. Here are some that may help the author to expand on:

  • Wellcome Global Monitor- First wave of findings. 2019
  • The next decades of vaccines: https://www.results.org.uk/sites/default/files/files/NextDecadeOfVaccines_Single_NoBleed.pdf
  • Hornsey et al 2018. Psychological roots of antivaccination attitudes
  • Larson et al 2016. The state of vaccine confidence – 67 country survey

Line 86: I would like to see more information on the screening process. The author indicated that initially a total of 178 records were identified, and following the screening process, a total of 30 articles were included. What did the screening process entail? What were the criteria for including/excluding articles?

Line 88: We need further information on this study and the methods used in this study. Although the authors cited their study, it is not yet peer-reviewed and published, so depending on the journal’s policy, it may need to be excluded from the references. Is there a preprint that we can read? We just don’t have any information on this study. Especially given that the author excluded unpublished records from their analysis (see Figure 1), it seems fair that this study is excluded as well. Alternatively, are there any published studies on vaccine acceptance rates in those two countries that the authors can include instead?

Figure 1: says full-text articles excluded for not meeting the eligibility criteria. The author should state explicitly what the eligibility criteria were.

Table 1: I am not able to double-check all the studies and the reported acceptance rates, but I spotted a mistake on the table, for the acceptance rates reported by Salali and Uysal 2020. The UK acceptance rate found in this study was 83% (as opposed to 86), and Turkey to %66 (as opposed to 69). The study found 3% of participants in both countries to refuse the vaccination, and this was not included in the reviewed calculation. I suggest the author check the reported acceptance rates in the studies with the ones on the table one more time, to make sure there are no further mistakes.   

Line 113: How are the country-specific acceptance rates calculated when there were multiple studies for the same country? Were there big differences in the reported acceptance rates for the same country? If so, it would be good to discuss why. There may be differences in the sampling methodology, which may lead to different acceptance rates for the same country.

Line 128: I’m not sure if it will be possible to reach a robust conclusion about changes in acceptance rates in a country if the studies were not longitudinal but cross-sectional studies done by different research groups. If those reported results are from different studies, the author should acknowledge this and discuss the possibility that the change in the acceptance rates may be merely because of the differences in sampling

Discussion:

It would be good to see a further discussion on why we are observing these differences among different countries, and whether the observed pattern for the COVID-19 vaccine acceptance rates matches with previous findings on vaccine hesitancy in those countries. From a quick glance, it looks like the patterns are similar. For example, recent reports (like the Wellcome global monitor) have shown lower acceptance rates in Eastern European countries, Russia and France, which also seems to be the case here. 

Finally, I think the article would benefit from a final proof-reading.

I hope these comments will be helpful in improving the manuscript, which I would like to see published. 

Author Response

Reviewer #2 comments

This study aims to compare the COVID-19 vaccine acceptance rates across countries by conducting a literature review on the recent survey studies conducted around the world.

I think this is an important review as it will allow researchers to observe the cross-cultural differences in COVID-19 vaccine acceptance rates and to investigate further why people in some countries are more hesitant about vaccinations compared to others. Therefore, I would like to see this article published. I have some concerns about the methodology, that I think we need further information on the article screening process to ensure the literature review was thorough and all eligible articles were included. Once we have more information on the eligibility criteria and an extended discussion and introduction (see details below), I think the manuscript will be suitable for publication.

Here are my further comments:

  1. Introduction: I would like to see more extended background information on the cross-cultural differences on vaccine hesitancy so that the author can also make predictions about which countries we would expect the COVID-19 vaccine acceptance rates would be lower. There are a few cross-cultural studies previously conducted on vaccine hesitancy worldwide, which should be cited. Here are some that may help the author to expand on:

Wellcome Global Monitor- First wave of findings. 2019

The next decades of vaccines: https://www.results.org.uk/sites/default/files/files/NextDecadeOfVaccines_Single_NoBleed.pdf

Hornsey et al 2018. Psychological roots of antivaccination attitudes

Larson et al 2016. The state of vaccine confidence – 67 country survey.

Response: I would like to thank the reviewer for the insightful comment that helped to clarify the major aim of this review; namely, to describe the variability in intent to get COVID-19 vaccines seen in different countries and regions globally. Based on the reviewer’s comments, the following paragraphs were added to the introduction section besides citing the suggested references:

(Page:2, lines: 51-79)

“Previous studies have shown that vaccine hesitancy is a common phenomenon globally, with variability in the cited reasons behind refusal of vaccine acceptance [11-13]. The most common reasons included: perceived risks vs. benefits, certain religious beliefs and lack of knowledge and awareness [14-16]. The aforementioned reasons can be applied to COVID-19 vaccine hesitancy, as shown by the recent publications that showed a strong correlation between intent to get coronavirus vaccines and its perceived safety [17], association of the negative attitude towards COVID-19 vaccines and unwillingness to get its vaccines [18], and the association of religiosity with lower intention to get COVID-19 vaccines [19].

Studying the global impact of vaccine hesitancy -including willingness to accept COVID-19 vaccines- could be complicated by the multifaceted nature of this phenomenon [1]. This entails the existence of cognitive, psychologic, socio-demographic and cultural factors that contribute to vaccine hesitancy [20-23]. Analysis of such factors is needed to address COVID-19 vaccine hesitancy, following the assessment of the scope and magnitude of this public health threat [24]. This can help in guiding interventional measures aimed to build and maintain response to tackle this threat [25].

Earlier studies that assessed attitude towards vaccines revealed the existence of regional variability in perceiving the safety and effectiveness of vaccination [12,26,27]. Higher-income regions were the least certain regarding vaccine safety with 72-73% of people in Northern America and Northern Europe who agreed that vaccines are safe. This rate was even lower in Western Europe (59%), and in Eastern Europe (50%), despite the presence of a substantial variability in Eastern European countries (from 32% in Ukraine, 48% in Russia, to 77% in Slovakia). On the other hand, the majority of people in lower-income areas agreed that vaccines are safe, with the highest proportions seen South Asia (95%) and in Eastern Africa (92%) [26]. A similar pattern was observed regarding vaccine effectiveness, with Eastern Europe as the region where people are the least likely to agree that vaccines are effective as opposed to South Asia and Eastern Africa [26]. The assessment of such regional differences can be invaluable in addressing and fighting public health threats posed by vaccine hesitancy [28].”

The references added:

  1. Lane, S.; MacDonald, N.E.; Marti, M.; Dumolard, L. Vaccine hesitancy around the globe: Analysis of three years of WHO/UNICEF Joint Reporting Form data-2015-2017. Vaccine 2018, 36, 3861-3867, doi:10.1016/j.vaccine.2018.03.063.
  2. Wagner, A.L.; Masters, N.B.; Domek, G.J.; Mathew, J.L.; Sun, X.; Asturias, E.J.; Ren, J.; Huang, Z.; Contreras-Roldan, I.L.; Gebremeskel, B., et al. Comparisons of Vaccine Hesitancy across Five Low- and Middle-Income Countries. Vaccines (Basel) 2019, 7, doi:10.3390/vaccines7040155.
  3. The Lancet Child & Adolescent Health. Vaccine hesitancy: a generation at risk. The Lancet. Child & adolescent health 2019, 3, 281, doi:10.1016/S2352-4642(19)30092-6.
  4. Karafillakis, E.; Larson, H.J.; consortium, A. The benefit of the doubt or doubts over benefits? A systematic literature review of perceived risks of vaccines in European populations. Vaccine 2017, 35, 4840-4850, doi:10.1016/j.vaccine.2017.07.061.
  5. Pelcic, G.; Karacic, S.; Mikirtichan, G.L.; Kubar, O.I.; Leavitt, F.J.; Cheng-Tek Tai, M.; Morishita, N.; Vuletic, S.; Tomasevic, L. Religious exception for vaccination or religious excuses for avoiding vaccination. Croat Med J 2016, 57, 516-521, doi:10.3325/cmj.2016.57.516.
  6. Yaqub, O.; Castle-Clarke, S.; Sevdalis, N.; Chataway, J. Attitudes to vaccination: a critical review. Soc Sci Med 2014, 112, 1-11, doi:10.1016/j.socscimed.2014.04.018.
  7. Karlsson, L.C.; Soveri, A.; Lewandowsky, S.; Karlsson, L.; Karlsson, H.; Nolvi, S.; Karukivi, M.; Lindfelt, M.; Antfolk, J. Fearing the disease or the vaccine: The case of COVID-19. Pers Individ Dif 2021, 172, 110590, doi:10.1016/j.paid.2020.110590.
  8. Paul, E.; Steptoe, A.; Fancourt, D. Attitudes towards vaccines and intention to vaccinate against COVID-19: Implications for public health communications. The Lancet Regional Health - Europe 2021, 1, doi:10.1016/j.lanepe.2020.100012.
  9. Olagoke, A.A.; Olagoke, O.O.; Hughes, A.M. Intention to Vaccinate Against the Novel 2019 Coronavirus Disease: The Role of Health Locus of Control and Religiosity. J Relig Health 2020, 10.1007/s10943-020-01090-9, doi:10.1007/s10943-020-01090-9.
  10. Murphy, J.; Vallieres, F.; Bentall, R.P.; Shevlin, M.; McBride, O.; Hartman, T.K.; McKay, R.; Bennett, K.; Mason, L.; Gibson-Miller, J., et al. Psychological characteristics associated with COVID-19 vaccine hesitancy and resistance in Ireland and the United Kingdom. Nat Commun 2021, 12, 29, doi:10.1038/s41467-020-20226-9.
  11. Pomares, T.D.; Buttenheim, A.M.; Amin, A.B.; Joyce, C.M.; Porter, R.M.; Bednarczyk, R.A.; Omer, S.B. Association of cognitive biases with human papillomavirus vaccine hesitancy: a cross-sectional study. Hum Vaccin Immunother 2020, 16, 1018-1023, doi:10.1080/21645515.2019.1698243.
  12. Browne, M.; Thomson, P.; Rockloff, M.J.; Pennycook, G. Going against the Herd: Psychological and Cultural Factors Underlying the 'Vaccination Confidence Gap'. PLoS One 2015, 10, e0132562, doi:10.1371/journal.pone.0132562.
  13. Hornsey, M.J.; Harris, E.A.; Fielding, K.S. The psychological roots of anti-vaccination attitudes: A 24-nation investigation. Health Psychol 2018, 37, 307-315, doi:10.1037/hea0000586.
  14. Lin, C.; Tu, P.; Beitsch, L.M. Confidence and Receptivity for COVID-19 Vaccines: A Rapid Systematic Review. Vaccines 2021, 9, 16.
  15. de Figueiredo, A.; Simas, C.; Karafillakis, E.; Paterson, P.; Larson, H.J. Mapping global trends in vaccine confidence and investigating barriers to vaccine uptake: a large-scale retrospective temporal modelling study. Lancet 2020, 396, 898-908, doi:10.1016/S0140-6736(20)31558-0.
  16. Wellcome Global Monitor. How does the world feel about science and health? Availabe online: https://wellcome.org/sites/default/files/wellcome-global-monitor-2018.pdf (accessed on 09-02-2021).
  17. Larson, H.J.; de Figueiredo, A.; Xiahong, Z.; Schulz, W.S.; Verger, P.; Johnston, I.G.; Cook, A.R.; Jones, N.S. The State of Vaccine Confidence 2016: Global Insights Through a 67-Country Survey. EBioMedicine 2016, 12, 295-301, doi:10.1016/j.ebiom.2016.08.042.
  18. The All-Party Parliamentary Group (APPG) on Vaccinations for All. The Next Decade of Vaccines: Addressing the challenges that remain towards achieving vaccinations for all. Availabe online: https://www.results.org.uk/sites/default/files/files/NextDecadeOfVaccines_Single_NoBleed.pdf (accessed on 09-02-2021).

  1. Line 86: I would like to see more information on the screening process. The author indicated that initially a total of 178 records were identified, and following the screening process, a total of 30 articles were included. What did the screening process entail? What were the criteria for including/excluding articles?

Response: I would like to thank the reviewer for this comment since it allowed the clarification of record screening process.

Based on the reviewer’s comment, the following paragraphs were added to the Material and Methods section:

(Page: 3, lines: 107-113)

“The inclusion criteria were: (1) Peer-reviewed published articles indexed in PubMed; (2) Survey studies among the general population, health-care workers, students, or parents/guardians); (3) The major aim of the study was to evaluate COVID-19 vaccine acceptance/hesitancy; (4) Publication language was English.

The exclusion criteria were: (1) Unpublished manuscripts (preprints); (2) The article did not aim to evaluate COVID-19 vaccine acceptance/hesitancy; (3) Publication language was not English.”

Since it appeared that (Figure 1) was confusing, the figure was replaced to indicate that initial identification resulted in 178 PubMed record retrieval, followed by title/abstract initial screening with exclusion of articles that did not tackle COVID-19 vaccine hesitancy/acceptance (n=125) and unpublished records in medrxiv (n=2).

This was followed by thorough screening of the articles (n=51), with exclusion of those that solely evaluated flu vaccine acceptance/hesitancy (n=3) and those that did not tackle COVID-19 vaccine hesitancy/acceptance (n=18).

You can find the updated (Figure 1) in Page 4 of the revised manuscript.

  1. Line 88: We need further information on this study and the methods used in this study. Although the authors cited their study, it is not yet peer-reviewed and published, so depending on the journal’s policy, it may need to be excluded from the references. Is there a preprint that we can read? We just don’t have any information on this study. Especially given that the author excluded unpublished records from their analysis (see Figure 1), it seems fair that this study is excluded as well. Alternatively, are there any published studies on vaccine acceptance rates in those two countries that the authors can include instead?

Response: I would like to thank the reviewer for this comment which is totally fair. Since this manuscript was accepted and published on January 12, 2021; I apologize for not updating the reference list and the parts of the manuscript referring to this study accordingly.

You can find the publication using the following link: https://www.mdpi.com/2076-393X/9/1/42

Also, I updated the reference list as follows:

  1. Sallam, M.; Dababseh, D.; Eid, H.; Al-Mahzoum, K.; Al-Haidar, A.; Taim, D.; Yaseen, A.; Ababneh, N.A.; Bakri, F.G.; Mahafzah, A. High rates of COVID-19 vaccine hesitancy and its association with conspiracy beliefs: A study in Jordan and Kuwait among other Arab countries. Vaccines (Basel) 2021, 9, 42, doi:10.3390/vaccines9010042.

I hope that this response would be sufficient to keep this article among the records in this review.

  1. Figure 1: says full-text articles excluded for not meeting the eligibility criteria. The author should state explicitly what the eligibility criteria were.  

Response: Based on the reviewer’s comment, the figure was updated (Page 4)

Also, please refer to response to comment no. 2

  1. Table 1: I am not able to double-check all the studies and the reported acceptance rates, but I spotted a mistake on the table, for the acceptance rates reported by Salali and Uysal 2020. The UK acceptance rate found in this study was 83% (as opposed to 86), and Turkey to %66 (as opposed to 69). The study found 3% of participants in both countries to refuse the vaccination, and this was not included in the reviewed calculation. I suggest the author check the reported acceptance rates in the studies with the ones on the table one more time, to make sure there are no further mistakes.

Response: I am deeply grateful for this meticulous comment from the reviewer, and based on the reviewer correction, the following corrections were made in the manuscript after double-checking the calculated rates:

Table 1: Rows 6, 11, 23 (UK), 27 (Turkey), 50 and 51

  1. Line 113: How are the country-specific acceptance rates calculated when there were multiple studies for the same country? Were there big differences in the reported acceptance rates for the same country? If so, it would be good to discuss why. There may be differences in the sampling methodology, which may lead to different acceptance rates for the same country.

Response: For multiple studies from the same country, a section in the results presented the changes only as a descriptive reporting of the different rates with time. I agree with the reviewer that different sampling strategies and possible biases in age, sex, socio-economic status of the enrolled participants might have resulted in the observed differences.  This was discussed in the limitations part of the Discussion section (Page 10, lines: 297-299).

For the graphical abstract, and the newly added Figure 2, the following paragraph was added to explain the basis of selection of the rate from countries with multiple studies: “For countries with more than one survey study, the vaccine acceptance rate of the latest survey was used in this graph. The estimates were also based on studies from the general population, except in the following cases were no studies from the general public were found (Australia: parents/guardians; DRC: healthcare workers; Hong Kong: healthcare workers; Malta: healthcare workers).”

  1. Line 128: I’m not sure if it will be possible to reach a robust conclusion about changes in acceptance rates in a country if the studies were not longitudinal but cross-sectional studies done by different research groups. If those reported results are from different studies, the author should acknowledge this and discuss the possibility that the change in the acceptance rates may be merely because of the differences in sampling.

Response: Based on the reviewer’s relevant comment, which I totally agree with, the following paragraph was added to the limitations part of the Discussion section: “In addition, the research studies included in this review represented cross-sectional studies, which can be seen as snapshots of vaccine hesitancy status in each country/region, with different sampling strategies, which may partly explain the differences in vaccine acceptance rates reported in various studies from a single country.”

  1. Discussion:

It would be good to see a further discussion on why we are observing these differences among different countries, and whether the observed pattern for the COVID-19 vaccine acceptance rates matches with previous findings on vaccine hesitancy in those countries. From a quick glance, it looks like the patterns are similar. For example, recent reports (like the Wellcome global monitor) have shown lower acceptance rates in Eastern European countries, Russia and France, which also seems to be the case here.

Response: I am deeply grateful for this helpful and insightful comment, and based on that the following paragraphs were added to the Discussion section:

Page 9, lines 237-239: “The relatively high rates of COVID-19 vaccine acceptance might be related to stronger confidence in vaccine safety and effectiveness, as reported previously in Asia”

Page 9, lines 250-252: “The results in Italy and France can be viewed from the perspective of lacking confidence in safety of these vaccines, since such a negative attitude was reported previously in these countries”

Page 9, lines 261-265: “Such patterns of COVID-19 vaccine hesitancy were consistent with a previous report which showed relatively high rates of vaccine hesitancy in Western and Eastern Europe, in addition to Russia [26]. The aforementioned low rates can be linked to lower confidence in vaccine safety and effectiveness in these regions”

  1. Finally, I think the article would benefit from a final proof-reading.

I hope these comments will be helpful in improving the manuscript, which I would like to see published.  

Response: I am deeply grateful for the meticulous, positive and constructive review of this manuscript by the reviewer, which helped in improving the manuscript to a huge degree.

Thanks a lot

Reviewer 3 Report

  1. The author is using the term vaccine acceptance when the appropriate term is vaccine acceptability.  Acceptance should only be used for actually getting the “jab” in their arm.
  2. The search is not broad enough; this is fundamentally a psychological and communication topic and the search should not be limited to Pubmed.  One example of a missed publication is a paper by Head and colleagues - https://doi.org/10.1177/1075547020960463
  3. The explanation about what went in and what was excluded is not adequate.  What were the articles that didn’t meet their criteria about; the criteria was not clearly specified. It seems inconsistent to exclude two papers because they are unpublished but to include your own unpublished data. 
  4. It appears that all papers were treated the same as if they measured vaccine acceptability the same way.  This is not always the case and can sometimes impact rates.  This variable should be discussed. 
  5. The finding that vaccine acceptability varied widely by country is interesting. It would be interesting to determine, to the extent possible, whether acceptability by age or sex was similar or different across countries. 

Author Response

Reviewer #3 comment

  1. The author is using the term vaccine acceptance when the appropriate term is vaccine acceptability. Acceptance should only be used for actually getting the “jab” in their arm.  

Response: I would like to thank the reviewer for this comment, and even though I agree with the reviewer that acceptability is a more accurate term to describe willingness to take the vaccine; the ubiquitous use of the term “vaccine acceptance” found in literature made me feel inclined to keep the term as it is

E.g.  Lazarus, J.V., Ratzan, S.C., Palayew, A. et al. A global survey of potential acceptance of a COVID-19 vaccine. Nat Med (2020).

  1. The search is not broad enough; this is fundamentally a psychological and communication topic and the search should not be limited to Pubmed. One example of a missed publication is a paper by Head and colleagues - https://doi.org/10.1177/1075547020960463.  

Response: I would like to thank the reviewer for this valuable comment; however, the major aim of this review was to provide a concise and succinct description of COVID-19 vaccine acceptance in different regions and countries globally, and to evaluate the regions where there is a gap of information about the scope of COVID-19 vaccine hesitancy. Thus, I would like to stick to the search in PubMed solely. However, based on the reviewer’s comment the following paragraph was added to the limitations part of the Discussion section (Page 10, lines 293-295): “This approach could have resulted in the inevitable missing of a few relevant studies tackling the subject of this review (e.g. the study by Head et al assessing SARS-CoV-2 vaccination intentions among adults in US) [103].”

  1. The explanation about what went in and what was excluded is not adequate. What were the articles that didn’t meet their criteria about; the criteria was not clearly specified. It seems inconsistent to exclude two papers because they are unpublished but to include your own unpublished data.  

Response: We would like to thank the reviewer for this comment that helped to clarify this some ambiguities in Materials and Methods section. Based on the reviewer comments, and the comment made kindly by reviewer no.2 as well, I changed Figure 1 to incorporate the reviewers’ valuable comments (please refer to Figure 1, page 4)

  1. It appears that all papers were treated the same as if they measured vaccine acceptability the same way. This is not always the case and can sometimes impact rates. This variable should be discussed.  

Response: I would like to thank the reviewer for this valuable and insightful comment. Based on the reviewer comment the following change was made to (Table 1) to incorporate the measurement of COVID-19 vaccine acceptance by various studies.

In Table 1, a fourth column was added “Response recorded as vaccine acceptance”

Also, in the Discussion section (the paragraph on limitations), the following paragraph was added: “Finally, an important limitation was related to the different approaches used to express the willingness to accept COVID-19 vaccines in various studies (i.e. some studies used a binary response of yes/no, while others used a scale of strongly agree/agree/neutral/disagree/strongly disagree to deduce the inclination towards vaccine acceptance, etc.); thus, this variable should be taken into account for accurate comparisons of vaccine acceptance rates between different studies.”

  1. Figure 1: says full-text articles excluded for not meeting the eligibility criteria. The author should state explicitly what the eligibility criteria were.  

Response: I would like to thank the reviewer for this important comment. The following paragraph was added to the Materials and Methods section, based on the reviewers’ comments (also, please refer to comment no. 2 of the second reviewer)

Page: 3, lines: 107-113

“The inclusion criteria were: (1) Peer-reviewed published articles indexed in PubMed; (2) Survey studies among the general population, health-care workers, students, or parents/guardians); (3) The major aim of the study was to evaluate COVID-19 vaccine acceptance/hesitancy; (4) Publication language was English.

The exclusion criteria were: (1) Unpublished manuscripts (preprints); (2) The article did not aim to evaluate COVID-19 vaccine acceptance/hesitancy; (3) Publication language was not English.”

  1. The finding that vaccine acceptability varied widely by country is interesting. It would be interesting to determine, to the extent possible, whether acceptability by age or sex was similar or different across countries.  

Response: I would like to thank the reviewer for this valuable comment. To include the point raised by the reviewer, I added another column to (Table 1) that assessed whether age/sex were correlated with higher rates of COVID-19 vaccine acceptability. The consistent pattern seen across 15 studies/countries was the finding that males were consistently more likely to show willingness to get COVID-19 vaccines if sex was significantly correlated with vaccine acceptability. Age was seen less frequently (11 studies/countries) as a factor when (in a few studies younger individuals were more willing to accept the vaccine, while in others older age was reported to be correlated with a higher willingness to get COVID-19 vaccines). However, large differences in age stratification between studies precluded further assessment of such a correlation.
Thus, based on the reviewer’s comment the following modifications were added to the manuscript:

In (Table 1), an eighth column was added “Age/Sex correlation with higher vaccine acceptance” that evaluated whether significant differences COVID-19 vaccine acceptability existed in relation to age/sex.

Additionally, in the Results section (Page 8, lines 184-186) the following paragraph was added: “Male sex was associated with significantly higher rates of COVID-19 vaccine in 15 countries/studies, while age was a significant factor in 11 studies/countries.”

Also, in the Discussion section (Page 10, lines 285-290), the following paragraph was added: “Finally, the assessment of sex and age role in vaccine hesitancy revealed that males were more inclined to accept COVID-19 vaccines. This can be related to their higher perception of COVID-19 dangers and lower belief in conspiratorial claims surrounding the disease [45,99,101]. These variables should be considered for interpretation of COVID-19 acceptance rates since sampling bias, particularly in sex distribution, can affect the reported rates.”